# Inverted-L Shaped Wideband MIMO Antenna for Millimeter-Wave 5G Applications

Amit Patel [1] , Alpesh Vala [1], Arpan Desai [1] , Issa Elfergani [2,3,*] , Hiren Mewada [4] , Keyur Mahant [1] , Chemseddine Zebiri [5] , Dharmendra Chauhan [1] and Jonathan Rodriguez [2]

1 Department of Electronics and Communication Engineering, Chandubhai S Patel Institute of Technology (CSPIT), Charotar University of Science and Technology (CHARUSAT), Changa 388421, India; amitvpatel.ec@charusat.ac.in (A.P.); alpeshvala.ec@charusat.ac.in (A.V.); arpandesai.ec@charusat.ac.in (A.D.); keyurmahant.ec@charusat.ac.in (K.M.); dharmendrachauhan.ec@charusat.ac.in (D.C.)
2 The Instituto de Telecomunicações, Campus Universitário de Santiago, 3810-193 Aveiro, Portugal; jonathan@av.it.pt
3 School of Engineering and Informatics, University of Bradford, Bradford BD7 1DP, UK
4 Department of Electrical Engineering, Prince Mohammad Bin Fahd University, Al Khobar 31952, Saudi Arabia; hmewada@pmu.edu.sa
5 Laboratoire d'Electronique de Puissance et Commande Industrielle (LEPCI), Department of Electronics, University of Ferhat Abbas, Sétif -1-, Sétif 19000, Algeria; czebiri@univ-setif.dz
* Correspondence: i.t.e.elfergani@av.it.pt or i.elfergani@bradford.ac.uk; Tel.: +35-12-3437-7900

**Abstract:** Interconnected three-element and four-element wideband MIMO antennas have been proposed for millimeter-wave 5G applications by performing numerical computations and carrying out experimental measurements. The antenna structure is realized using Rogers 5880 substrate ($\varepsilon_r$ = 2.2, tan δ = 0.0009), where the radiating element has the shape of an inverted L with a partial ground. The unit element is carefully designed and positioned (by orthogonally rotating the elements) to form three-element (case 1) and four-element (case 2) MIMO antennas. The interconnected ground for both cases is ascertained to increase the practical utilization of the resonator. The proposed MIMO antenna size is (0.95λ × 3λ) for case 1 and (2.01λ × 1.95λ) for case 2 (at the lowest functional frequency). Both the designs give an impedance bandwidth of approximately 26–40 GHz (43%). Moreover, they achieve greater than 15 dB isolation and more than 6 dBi gain with an ECC value lower than 0.02, which meets the MIMO diversity performance thus making the three-element and four-element MIMO antennas the best choice for millimeter-wave 5G applications.

**Keywords:** connected ground; mm wave; MIMO; envelope correlation coefficient (ECC); antenna; inverted L antenna (ILA)

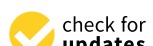



## 1. Introduction

Wireless communication has led to significant improvements in human life. Some of the limiting factors for communication are spectrum availability, power limitation, and channel fading. However, the demand for higher data rates for future-generation wireless technology (5th Generation (5G) Technology) is ever increasing. To run high bandwidth applications over an existing cellular network, the latest infrastructure development is needed for next-generation technology [1]. Presently, the lower frequency band (less than 6 GHz) has been already utilized, including the 5G NR sub −6 GHz [2–4]. For higher data rates such as gigabits per second, the 5G millimeter-wave band is in development, such as 5G new radio n257/n258/n260/n261. Many countries are developing 5G millimeter bands; for example, 28 GHz in Korea, 27.5 to 28.8 GHz in Japan, 24.25 to 27.5 and 37 GHz to 43.5 GHz in China, and 28 GHz, 37 GHz, and 39 GHz in the USA [5].

One of the available technologies to overcome the challenges and fulfill the demands is UWB (ultra-wideband)-MIMO systems [5]. UWB wireless technology fulfills the requirement of wide bandwidth and high gain, but its performance is limited in the multipath

wireless environment. The possible solution for that is UWB with MIMO wireless communication systems i.e., the use of multiple antennas at the transceiver. MIMO wireless communication systems enhance the performance in multipath propagation [6]. 5G technology uses the concept of massive MIMO, i.e., hundreds of elements are used simultaneously as transceivers [7]. The data rate of an MIMO system is directly proportional to the number of antennas used in the system [8]. Furthermore, as we increase the number of antennas in the MIMO systems, this will proportionally increase the data rate and channel capacity [9]. However, it will also reduce the space between the antennas, with the increase in the number of antennas resulting in an increase in the spatial correlation between the two received signals/antennas [10]. As the spatial correlation increases, the error rate performance decreases [11]. Hence, to improve the error rate performance of MIMO wireless communication systems, the primary requirements for MIMO systems are low spatial correlation and high isolation between the antennas [12]. However, achieving the required isolation levels between antennas with connected ground structures is a very challenging task [13,14]. In [15], defected ground structure (DGS) with an antenna is used to improve the isolation between the antennas in MIMO systems. Similarly, other techniques have been proposed, such as electronic band gap (EBG) [16] and split ring resonator (SRR) [17].

Various configurations of the MIMO antenna have been proposed: two-element MIMO resonating at a single band with a connected ground plane [18], dual-band two-element MIMO antenna with a separate ground plane [19]; three-element MIMO antenna with pattern and polarization diversity [20]; and various four-element MIMO antennas operating at single band [21,22], dual band [23–25], and wideband [26,27].

Dual-frequency operation for 5G applications is proposed in [23,24], however, it is does not cover the entire 5G spectrum. The proposed work in this article covers wideband frequency from 26–40 GHz, covering most of the millimeter-wave 5G frequency spectrum. A wideband antenna for 5G application is proposed in [25,26]; however, unlike the proposed work, the size is a limiting factor. A compact wideband MIMO antenna is proposed in [27] covering the entire 5G millimeter-wave band [27]; however, the separate ground structure makes the practical utilization of the antenna limited.

A four-element wideband MIMO antenna with 9.33 GHz at the center frequency is proposed in [28]. It covers the frequency range from 2.77 GHz to 12 GHz. A fractal singular ring is used as the front side of the antenna and a trapezoidal partial ground is at the bottom plane of the antenna. In [29], a MIMO antenna for 5G handheld devices is proposed and implemented. A total of six antennas are part of the design, out of which two are working on a sub-6 GHz band and four are working on millimeter-wave 5G band. A dual-band antenna with two elements at the operating frequencies of 2.4 GHz and 5.8 GHz is proposed in [30]. To improve the isolation between the elements, adopting parasitic elements and a defected ground plane is used. A tree-shaped planar MIMO antenna for 5G millimeter-wave applications is proposed in [31]. To achieve wideband response, radiating elements with various arcs are used. A small-slot MIMO antenna that can be embedded in smart glasses is designed and experimentally validated in [32].

In this paper, wideband three-element and four-element-connected ground MIMO antenna are proposed and realized for the 26–40 GHz frequency band. Initially, single-element design has been optimized for the wideband frequency resonance, and with the use of this single-element structure, three-element and four-element MIMO antennas are realized. Both the MIMO antennas have a connected ground plane that meets the prime requirement for the usage of the antenna in practical applications. The proposed design provides more than 20 dB of isolation with the connected ground plane design as well as good diversity parameters, with an ECC value lower than 0.02 and DG greater than 9.9.

The structure of this article is as follows. Unit element design and its simulation results are given in Section 1. Three-element design and its simulation result are covered in Section 2. Section 3 includes the design and analysis of the four-element MIMO antenna design. Prototype realization and the measurement results of three-element and four-

element antenna are discussed in Section 4. MIMO diversity analysis is included in Section 5, followed by the conclusions.

## 2. Unit Element

The unit element design of the resonating structure is illustrated in Figure 1. An inverted L-shaped antenna (ILA) structure with truncated ground and a size of 15 × 15 mm$^2$ is proposed. The detailed dimension of the antenna is depicted in Figure 1. RT Duroid 5880 material ($\varepsilon r = 2.2$, and tan $\delta = 0.0009$) is used as substrate, with a height of 1.6 mm.

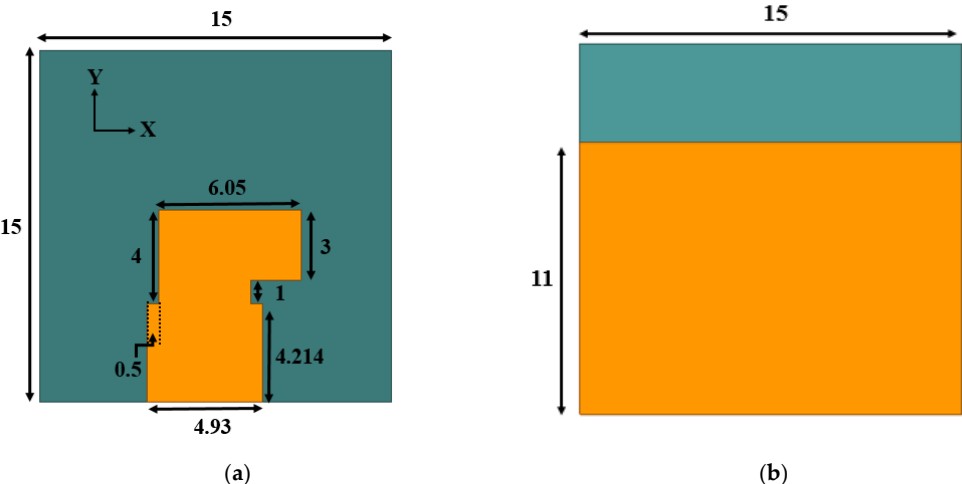

(a)  (b)

**Figure 1.** Unit element antenna geometry with truncated ground: (**a**) front side; (**b**) back side.

The proposed structure has been simulated using computer simulation technology (CST) software. Initially, the ground plane is kept without any truncation for the simulation of the element. However, the proposed antenna did not give the expected result; hence, we have modified the ground plane such that it improves the operating bandwidth of the antenna. Parametric analysis has been performed by varying the length of the ground plane to achieve the wideband response.

The result of the reflection coefficient value for the different ground planes is shown in Figure 2. The length of the ground plane is tuned from 7 mm to 15 mm (full ground plane). The result indicates that a ground length of 11 mm provides a better reflection coefficient and impedance bandwidth in the operating range of (43%) 26–40 GHz compared to a full ground plane antenna, which provides an impedance bandwidth of (33%) 27–38 GHz.

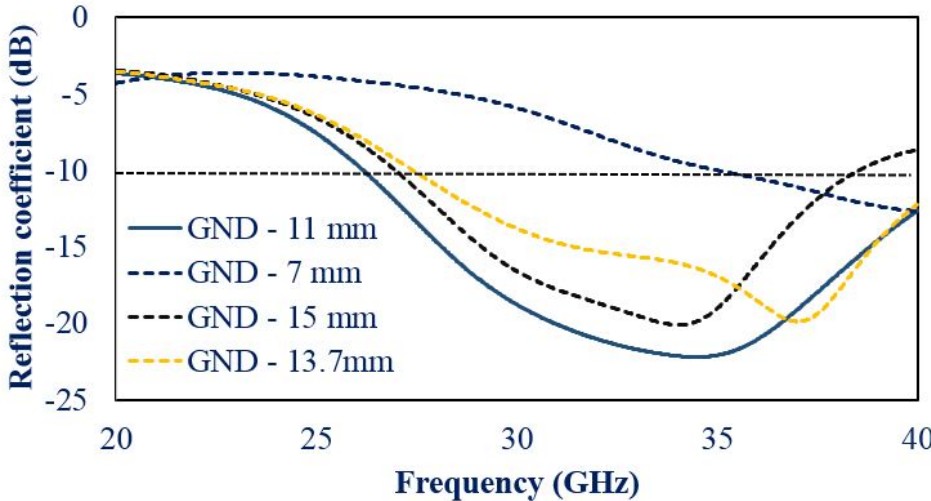

**Figure 2.** Reflection coefficient values for the various ground plane size.

## 3. Element MIMO Design (Case 1)

The unit element finalized in the last section is used to realize the three-element MIMO antenna. To finalize the geometry (placement of element) of the MIMO antenna, different configurations have been simulated for the isolation. Figure 3 shows the three-element connected ground antenna geometry chosen after verifying the optimum results of the MIMO antenna from Figure 2. The finalized physical structure of the three-elements MIMO antenna provides the maximum isolation between them. Elements of the antenna and slots on the ground plane are carefully optimized to ensure spatial diversity and thus achieve maximum isolation. Here, port 2 and port 3 elements are placed orthogonally to the port 1 element. Two vertical slots are created in the bottom plane for enhancing the isolation performance while still ensuring that the ground is connected.

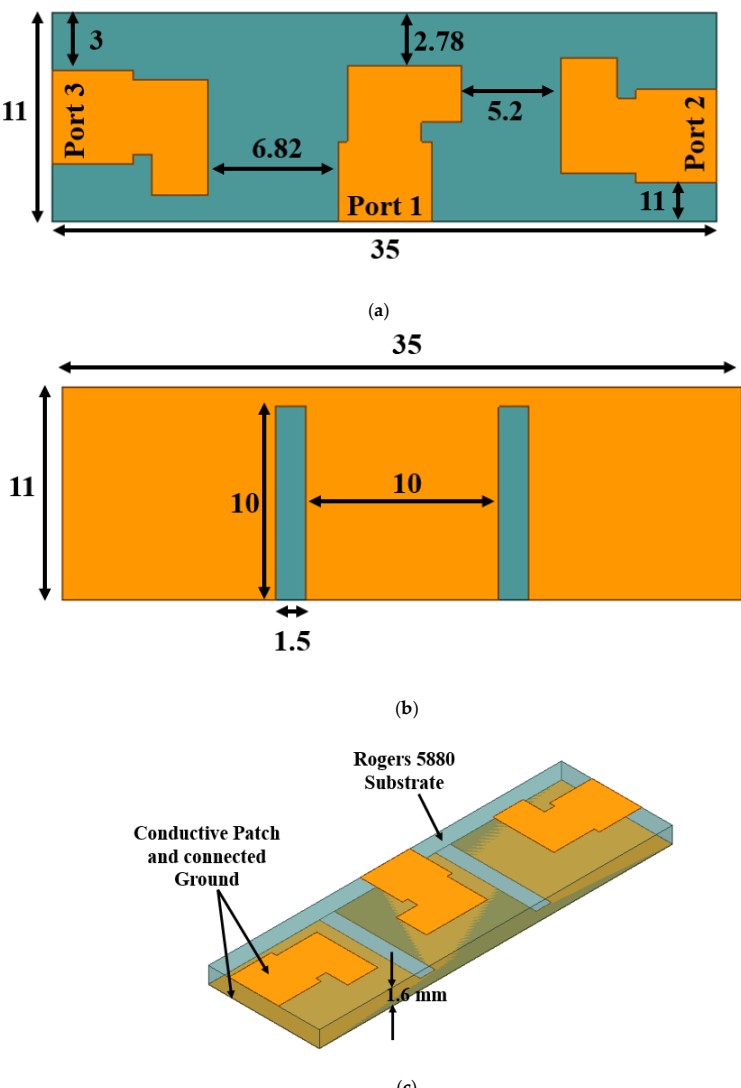

**Figure 3.** Three-element MIMO antenna geometry: (**a**) top plane; (**b**) bottom plane; (**c**) 3D view.

The size of the three-element MIMO antenna is $11 \times 35$ mm$^2$. The simulation result for the S-parameters of the design is shown in Figure 4. In MIMO antenna applications, there is expected to be maximum isolation between the elements when they are transmitting signals. Mutual coupling arises due to the induction of voltage in one element because of current induction from another element. It is a prerequisite that there should be maximum isolation among all the elements of MIMO. It is observed that the isolation level is more than 20 dB with an impedance bandwidth ranging between (43%) 26 GHz to

40 GHz. This shows that good isolation between the elements is achieved without adding a complex geometry.

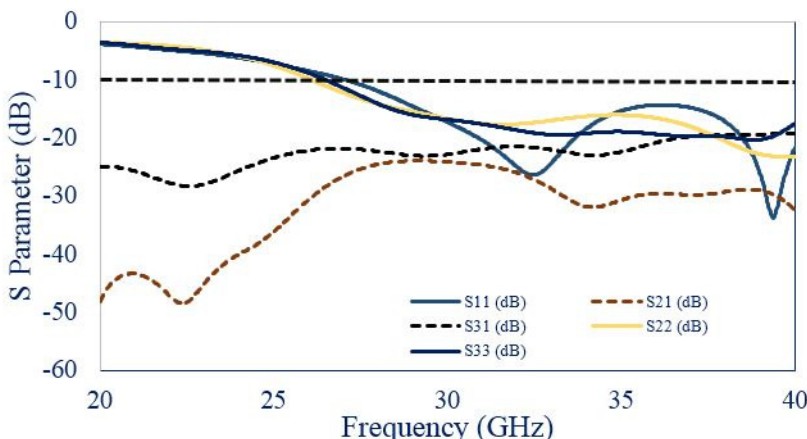

**Figure 4.** S-Parameters of three-element MIMO Antenna.

To further analyze the coupling amongst the elements of the MIMO antenna, surface current distribution has been examined as depicted in Figure 5 at 26 GHz, 30 GHz, and 35 GHz by exciting port 1 and terminating other ports at a matched load (50 Ω). Minor current coupling is observed between elements due to the chosen antenna geometry.

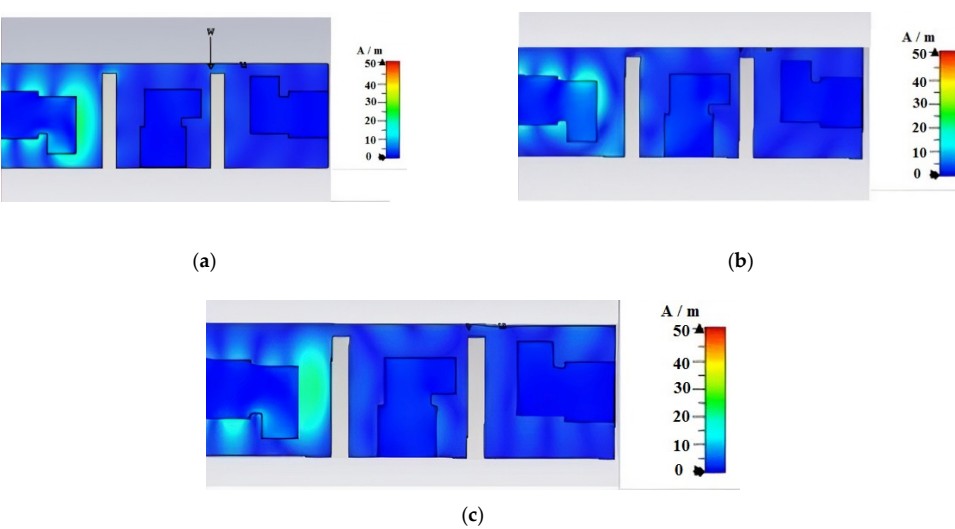

**Figure 5.** The flow of current on the surface by exciting port 1 and keeping other ports terminated with a match load (**a**) 26 GHz (**b**) 30 GHz (**c**) 35 GHz.

## 4. Element MIMO Design (Case 2)

As a greater number of elements are preferred for accommodating a larger number of users, a four-element MIMO antenna design is accomplished using by arranging the unit cell element in an orthogonal manner with carefully engineered connected ground geometry. As discussed earlier the important parameter is the isolation between the elements of the MIMO antenna. To determine the better isolation among the elements, various configuration have been studied and simulated using CST software. To obtain better isolation, the element position was changed between the different configurations shown in Figure 6a (case-I) and Figure 6b (case-II).

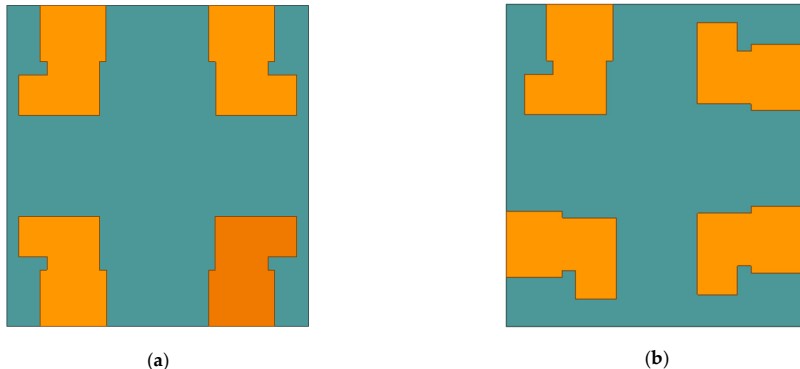

**Figure 6.** Four-element configuration variations for better isolation (**a**) case-I and (**b**) case-II.

Simulation was conducted to determine the transmission coefficient of the ports of the MIMO antenna, and the simulation result for design 1 (Figure 6a) and design 2 (Figure 6b) is shown in Figure 7. The simulation result for configuration 1 for the transmission coefficient is shown in Figure 7a. This shows that the isolation between first and second elements, as well as third and fourth elements, is less than 10 dB from the frequency range of 34 GHz to 40 GHz. To enhance the isolation, the position of the fourth element is changed as shown in Figure 6b. The position of the fourth element is changed, and the corresponding simulation result for the antenna is depicted in Figure 7b.

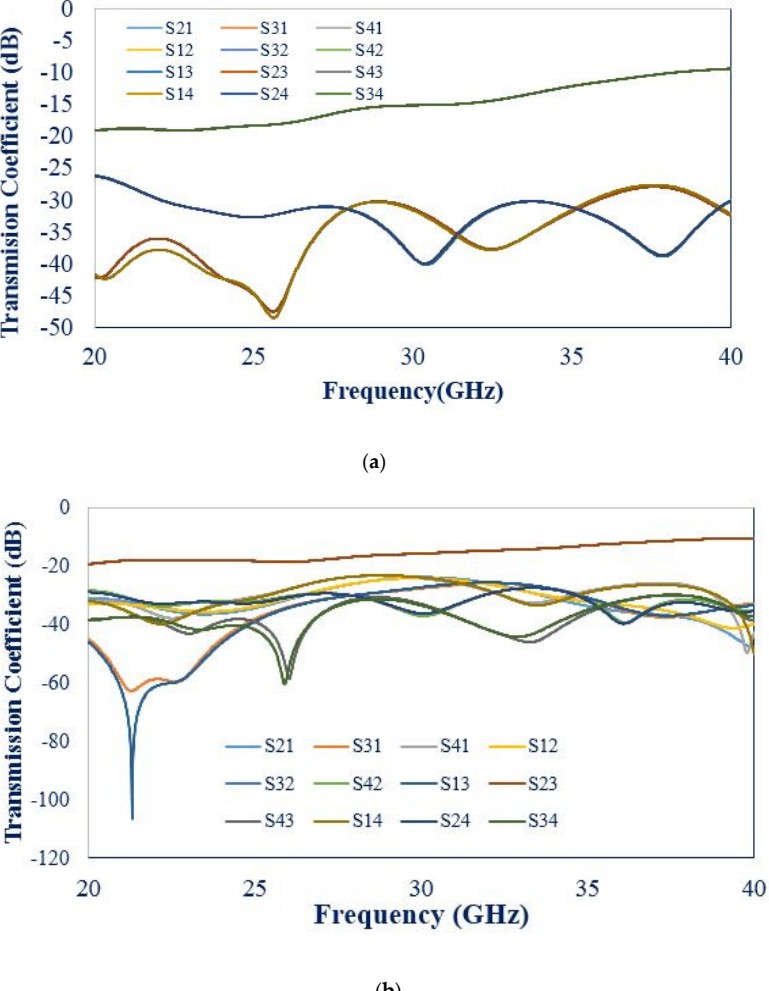

**Figure 7.** Transmission coefficient of (**a**) Figure 6a (case-I); (**b**) Figure 6b (case-II).

This indicates that the isolation between the second port and third port is better than 15 dB for the frequency range of 30 GHz to 40 GHz. Further modifications are made to the structure as shown in Figure 8, which improved the isolation. These provided better results and a detailed description is given below.

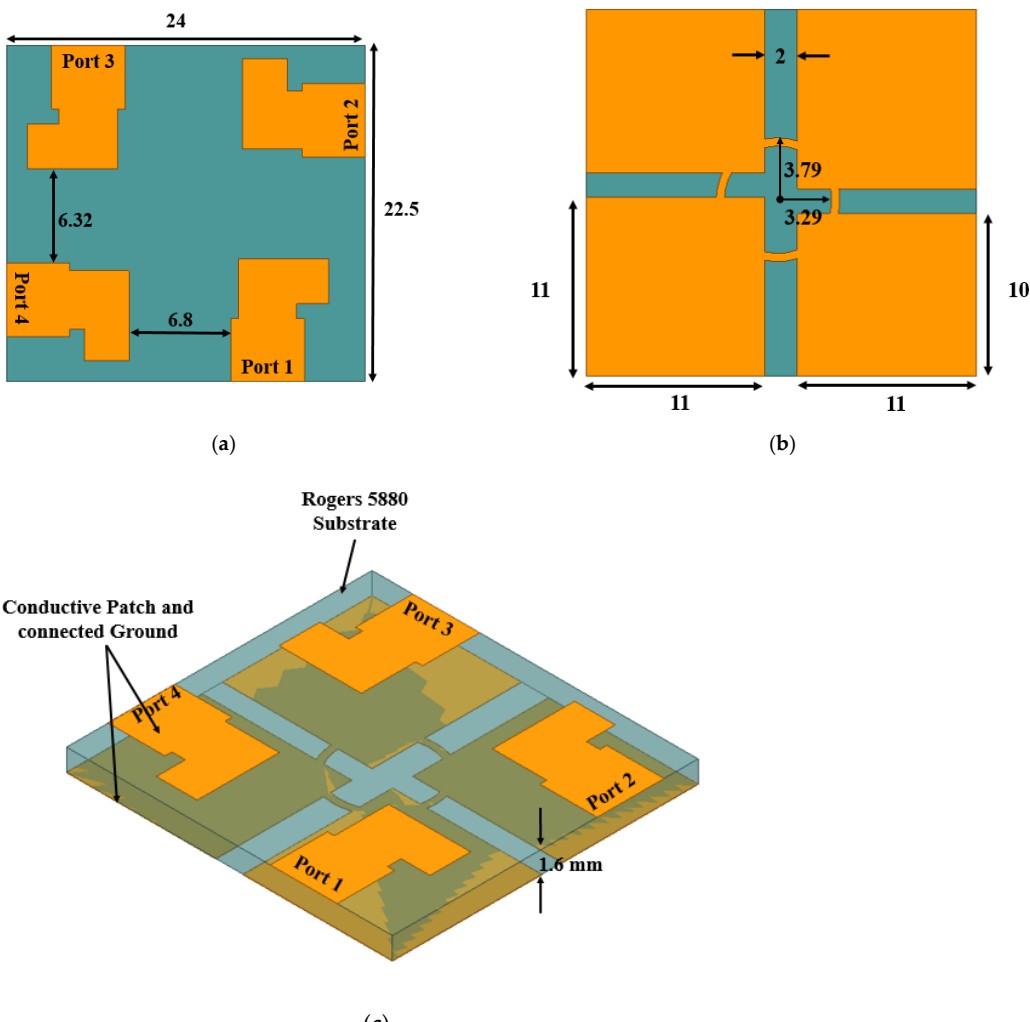

**Figure 8.** Four-element MIMO antenna: (**a**) top plane; (**b**) bottom plane; and (**c**) side view.

Figure 8 shows the chosen four-element MIMO antenna geometry after analyzing the results in terms of S-parameters. The antenna elements are arranged in an orthogonal rotational manner as illustrated in Figure 8a. As shown in Figure 8b, the ground plane is interconnected using a circular ring connecting a slotted plus shape. Figure 8c presents the 3D view of the proposed design.

The simulated reflection coefficient is shown in Figure 9. It shows a passband response from 26 GHz to 40 GHz. Transmission coefficient values between all the ports are depicted in Figure 10. The achieved isolation amongst all the ports is better than 24 dB.

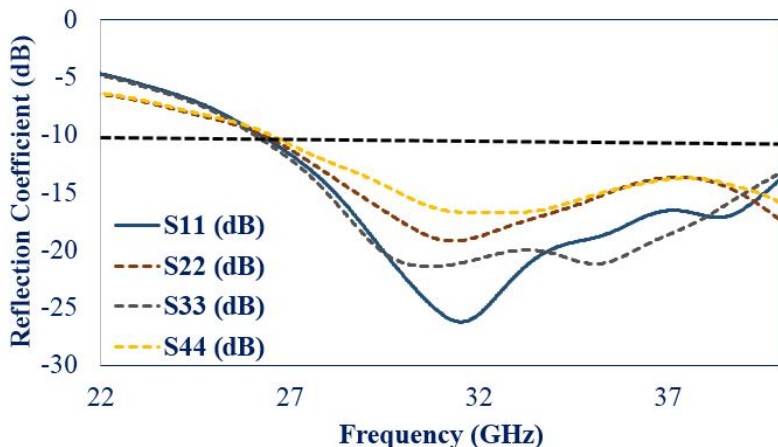

**Figure 9.** Reflection coefficient results for four-port MIMO antenna.

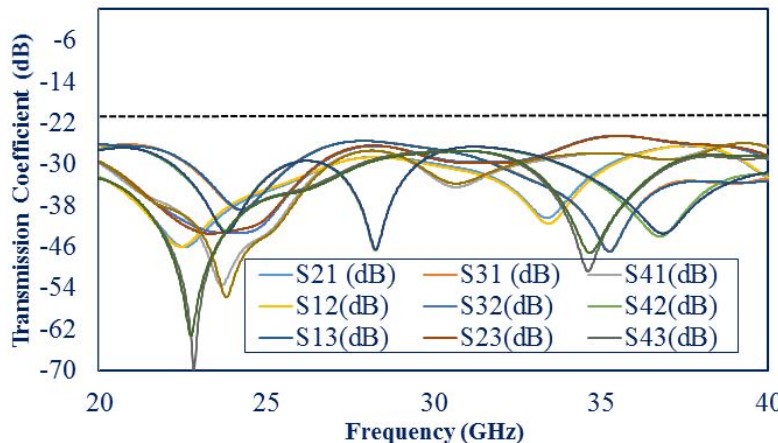

**Figure 10.** Simulated transmission coefficient results for four-port MIMO antenna.

To examine the leakage of the current between the inter-elements, the current distribution is analyzed as shown in Figure 11. Excitation is given to individual elements and the respective current linkage to other elements is simulated by terminating other element ports with a matched load. The result shows that a minor current is linked between ports and the elements with the connected ground.

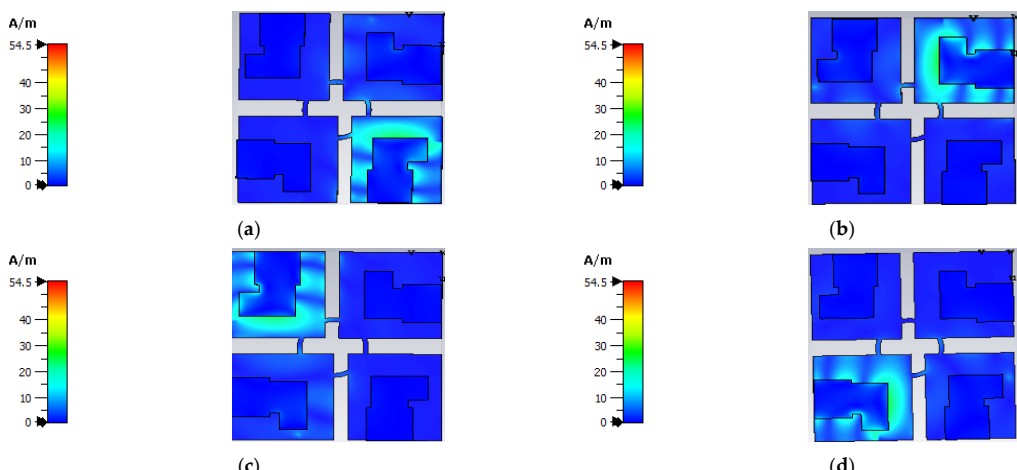

**Figure 11.** Surface current distribution when excitation is given to (**a**) port 1, (**b**) port 2, (**c**), port 3, and (**d**) port 4.

The 3D radiation pattern is plotted as depicted in Figure 12 for element 1 by terminating all the ports with a matched load. It shows that the antenna provides gain of 8.48 dBi, 9.49 dBi, and 10.2 dBi at 26 GHz, 30 GHz, and 35 GHz, respectively.

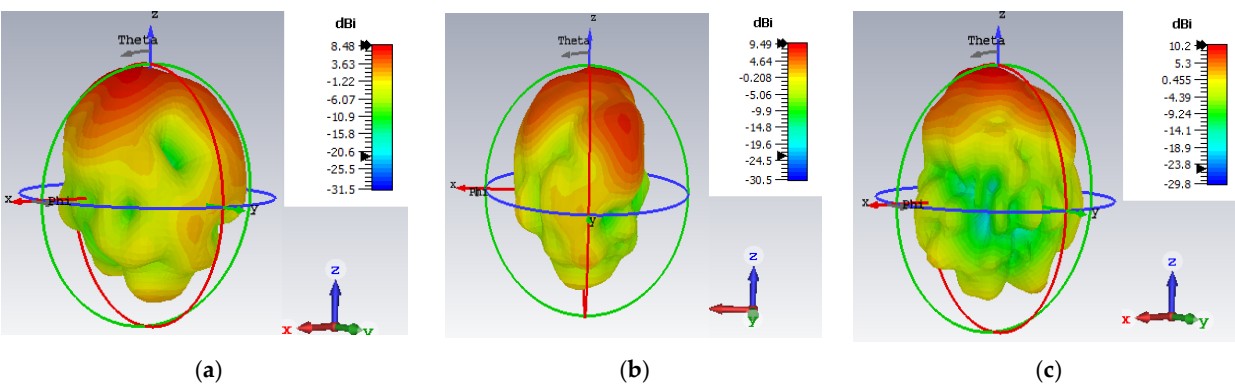

**Figure 12.** 3D radiation of element 1 at (**a**) 26 GHz, (**b**) 30 GHz, and (**c**) 35 GHz.

## 5. Fabrication and Measurement of Case 1 (Three-Element) and Case 2 (Four-Element) Connected Ground Antenna

The prototype realization for the case 1 (three-element) and case 2 (four-element) MIMO antenna is depicted in Figure 13, respectively. It is realized on RT duroid 5880 substrates (thickness = 1.6 mm). Measurements are performed for both cases.

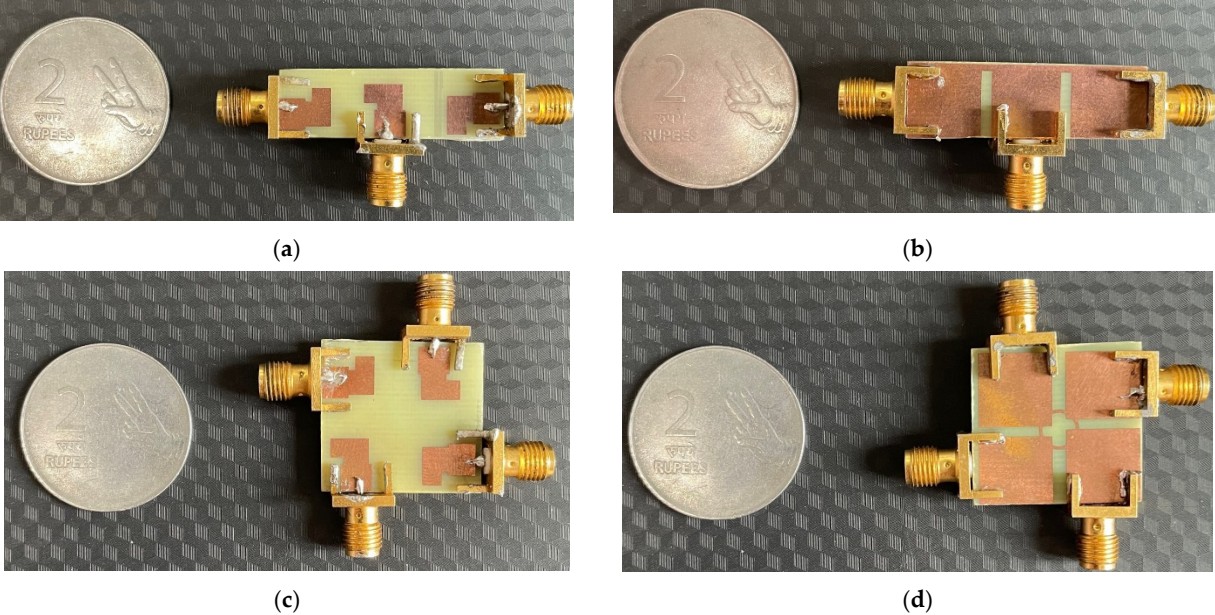

**Figure 13.** Prototype realization. (**a**) Three-element MIMO antenna—top view; (**b**) three-element MIMO antenna—bottom view; (**c**) four-element MIMO antenna—top view; (**d**) four-element MIMO antenna—bottom view.

The measured results of both cases are shown in Figures 14 and 15, respectively. The impedance bandwidth of the antenna with case 1 (three-element) spans between (42.5%), whereas for the antenna with case 2 (four-element) it ranges between (43%). The isolation in both cases is more than 24 dB. The experimental and simulated results show decent similarity with minor variations because of connector losses and fabrication error.

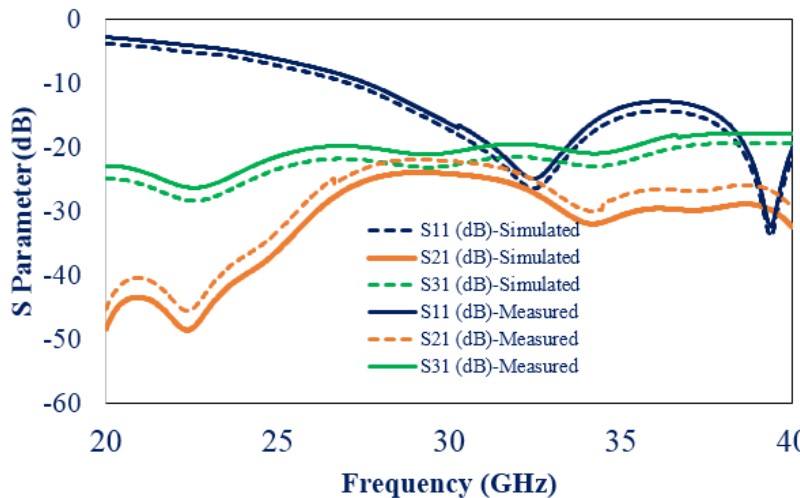

**Figure 14.** Measured S parameter result of three-element antenna (case 1).

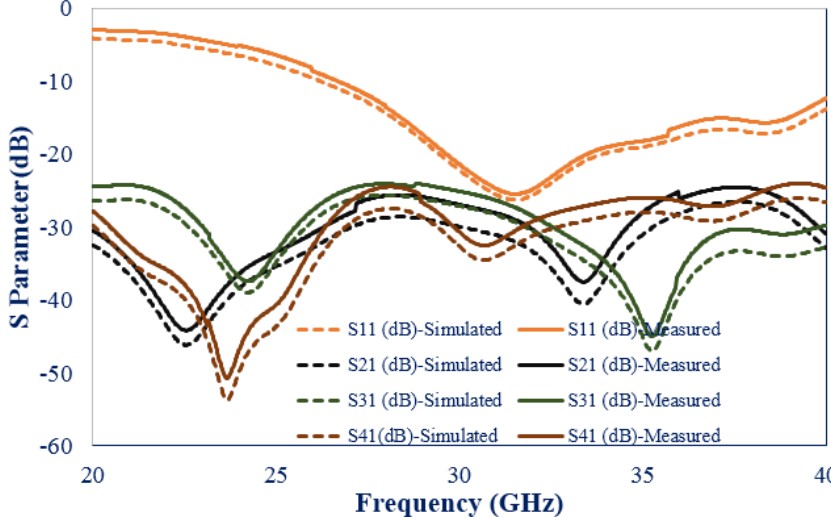

**Figure 15.** Measured S parameter results of the four-element antenna (case 2).

Figure 16 shows the radiation measurement setup in an anechoic chamber where the three-element or four-element MIMO antenna is placed along the E and H planes for the radiation pattern measurement. The co-cross component results shows that separation of more than 10 dB is achieved for both cases, as shown in Figure 17a–d, where the E and H plane pattern shows an omnidirectional pattern with a null on the lower side upon placing the antenna along the H plane.

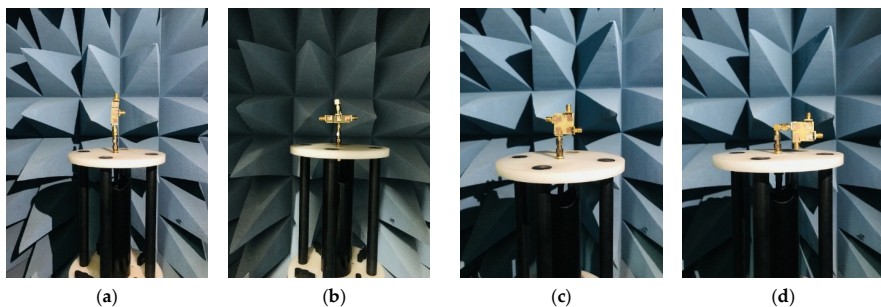

**Figure 16.** Radiation pattern measurement setup in anechoic chamber: (**a**) three-element E plane; (**b**) three-element H plane; (**c**) four-element E plane; (**d**) four-element H plane.

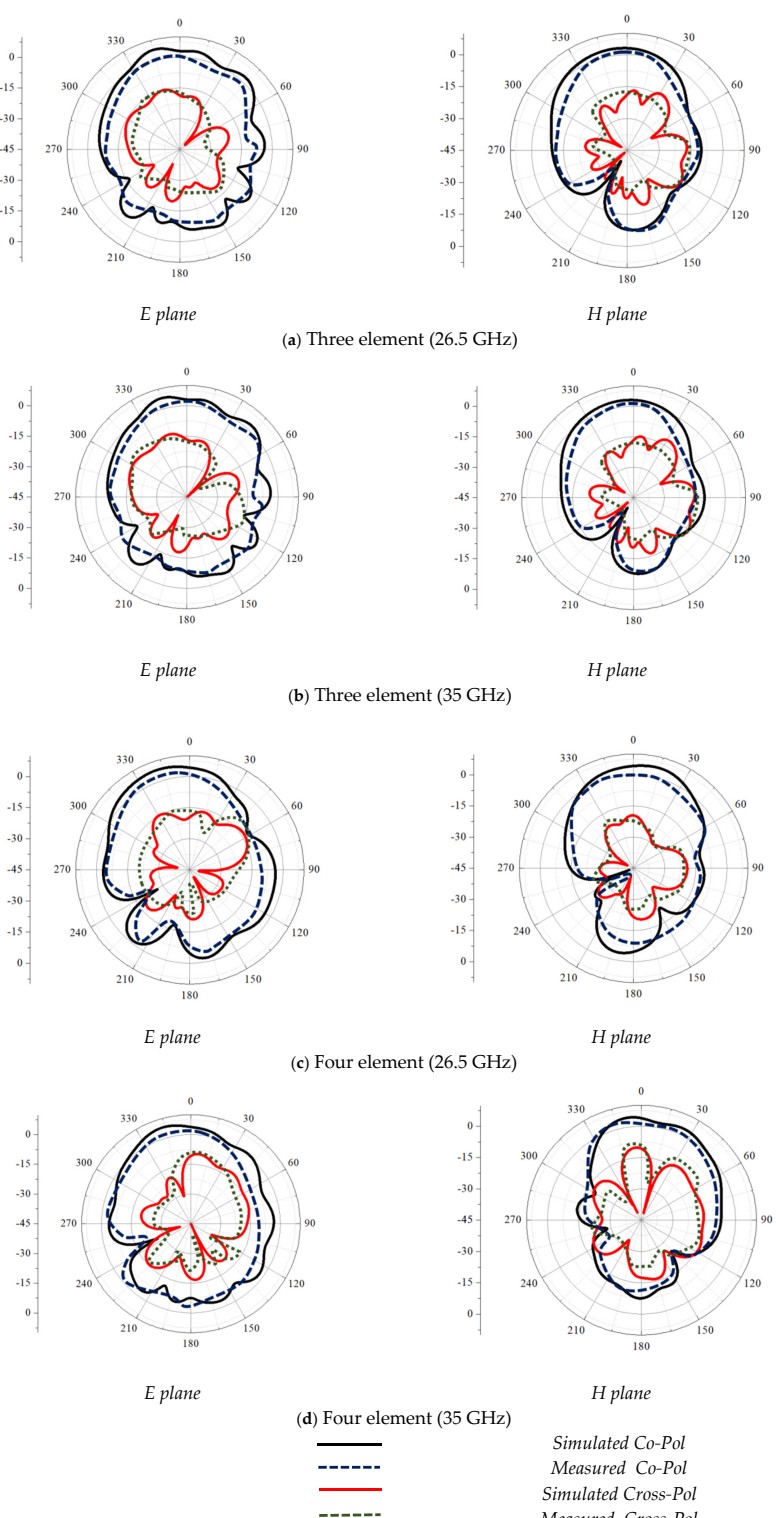

E plane　　　　　　　　　　　　H plane

(**a**) Three element (26.5 GHz)

E plane　　　　　　　　　　　　H plane

(**b**) Three element (35 GHz)

E plane　　　　　　　　　　　　H plane

(**c**) Four element (26.5 GHz)

E plane　　　　　　　　　　　　H plane

(**d**) Four element (35 GHz)

*Simulated Co-Pol*
*Measured Co-Pol*
*Simulated Cross-Pol*
*Measured Cross-Pol*

**Figure 17.** Simulated and measured co-pol and cross-pol patterns of the antenna.

## 6. MIMO Diversity Analysis

The practical utilization of the MIMO antenna needs careful analysis of diversity performance. The important parameters that need to be addressed include ECC and DG.

The evaluation of diversity capability is performed using ECC for a system with multiple antennas. The resemblance among the received signals is evaluated using the ECC parameter. A lower ECC indicates lower mutual coupling where the safe threshold is 1. ECC value should be between 0 and 1. The ECC value should be evaluated using

far-field parameters as suggested in [14]. The value of ECC can be calculated by referring to Equation (1).

$$\rho_e = \frac{\left| \underset{4\pi}{\iint} [s_i(\theta, \phi)] * [s_i(\theta, \phi)] d\Omega \right|}{\underset{4\pi}{\iint} |s_i(\theta, \phi)|^2 d\Omega \underset{4\pi}{\iint} |s_i(\theta, \phi)|^2 d\Omega} \tag{1}$$

where $\rho_e$ is the 3D radiation port of the MIMO antenna when excitation is given to the *i*th port. The asterisk indicates the Hermitian operator in a form of the product. Here for the measurement, the outdoor environment is considered.

Figure 18 indicates the simulated and measured ECC and DG values of the proposed element. The value of the ECC is less than 0.02 and DG > 9.9, which satisfied the criteria for the diversity performance.

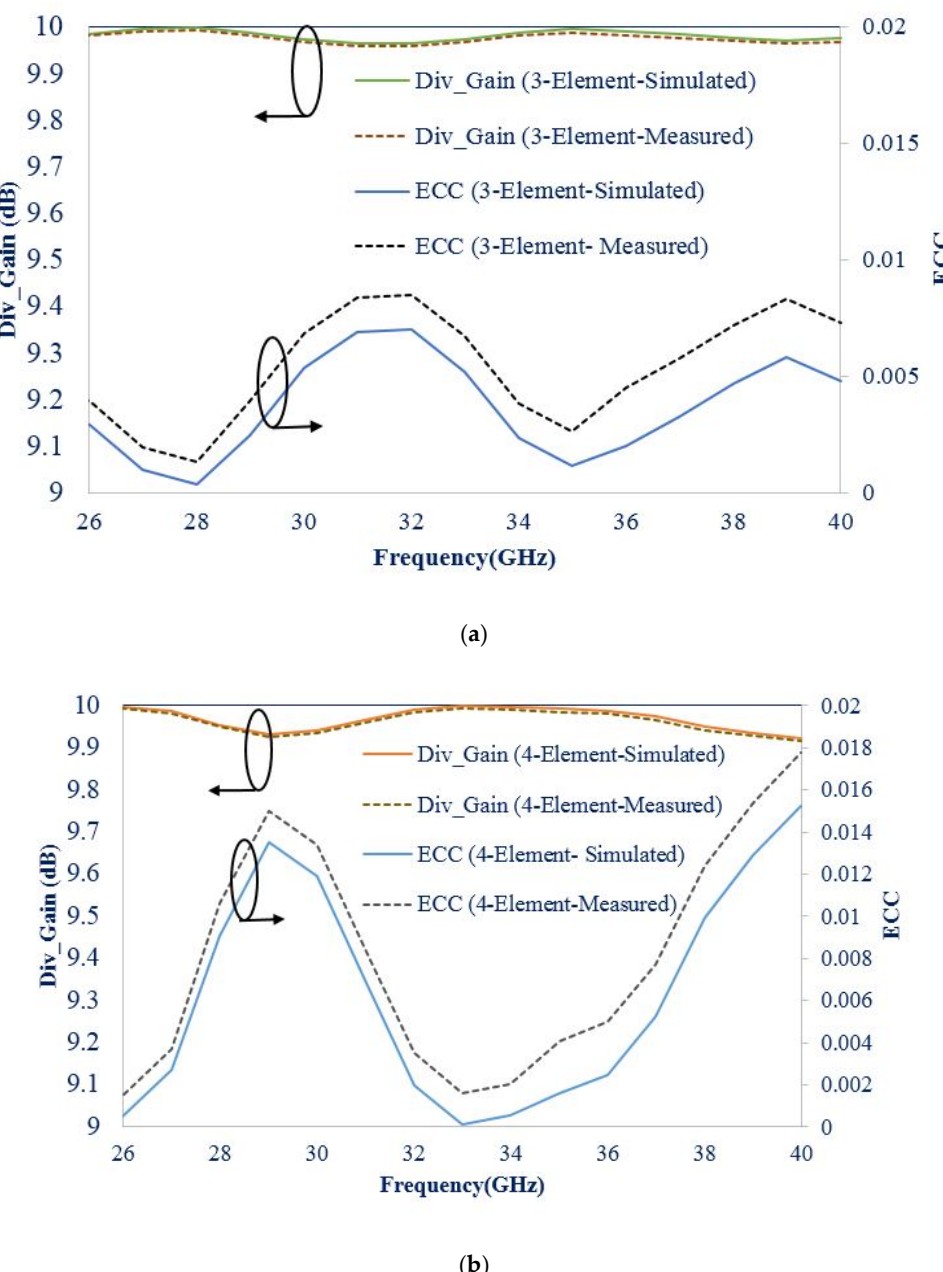

(a)

(b)

**Figure 18.** DG and ECC value of the proposed antenna: (**a**) three elements; (**b**) four elements.

The results for the efficiency and gain for both cases are shown in Figure 19. The result indicates more than 5 dBi gain for the three-elements and more than 7 dBi gain for the four-element MIMO antenna. The efficiency of the element is more than 70% for the band of interest in both cases.

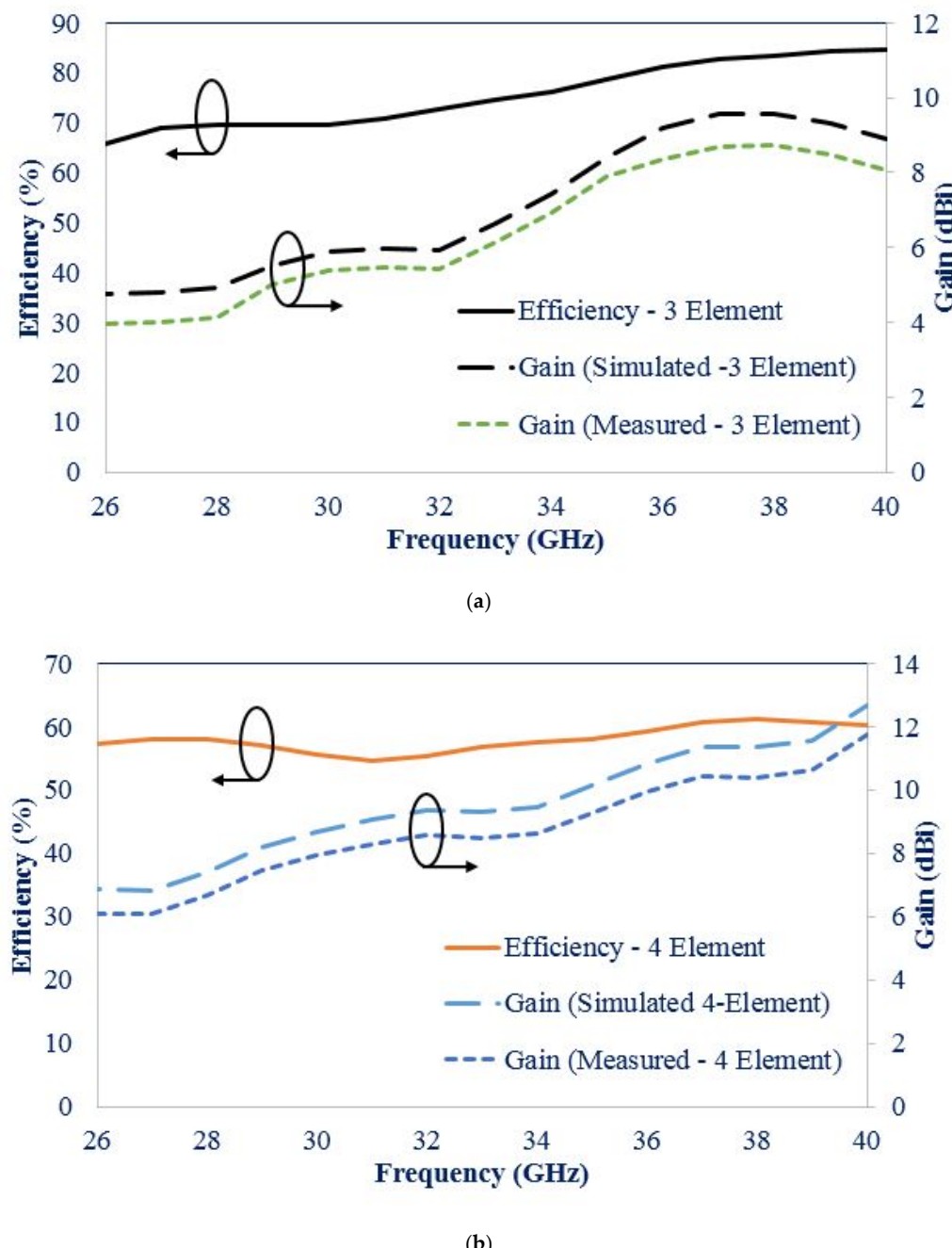

(**a**)

(**b**)

**Figure 19.** Efficiency and gain of the proposed (**a**) three-element and (**b**) four-element MIMO antenna.

A comparison has been conducted for proposed three- and four-element MIMO antennas with previously published work. The parameters that are used for the comparison are material usage, elements, gain, efficiency, isolation, and ECC. A proposed design provides a wideband response with better MIMO diversity. Isolation between the ports is better than 20 dB. ECC values are less than 0.01 for the three-element antenna and 0.02 for the four-element antenna.

From Table 1, it can be observed that [20,21] provides single-frequency operation whereas the proposed antenna provides wideband frequency response with compact size, better ECC, and gain value. The antenna proposed in [22,23] provides a dual-frequency response, which is overcome in the proposed design as it operates in the wideband regime with a compact size. The size of the proposed design is small compared to [24,25], with a wider frequency response and better isolation between the ports. The MIMO antenna in [26] has a compact size compared to the proposed antenna; however, the separate ground structure makes the practical utilization of the antenna very limited.

**Table 1.** Comparison of proposed three-element (case 1) and four-element (case 2) with other MIMO antennas.

| Ref | Frequency (GHz) | Ports/Ground | Unit Cell Dimension (mm²) | Gain (dBi) | Efficiency (%) | Isolation (dB) | ECC | Material |
|---|---|---|---|---|---|---|---|---|
| [18] | 28 | Two/Connected | 33 × 27.5 | 6.9 | - | >30 | − | TLY-5 |
| [19] | 26.65–29.2; 36–42 | Two/Separate | 26 × 11 | 5 | 99.5, 98.6 | >25 | <0.002 | Rogers 5880 |
| [21] | 28 | Four/Connected | 30 × 30 | 6.1 | 92 | 29 | <0.16 | Rogers 5880 |
| [20] | 28 | Four/Connected | 48 × 31 | 10 | - | >21 | <0.0015 | Neltec |
| [22] | 27.6–28.6; 37.4–38.6 | Four/Connected | 20 × 24 | 7.9 | >85 | >28 | <0.001 | Rogers 5880 |
| [23] | 24.10–27.18; 33–44.13 | Four/Connected | 24 × 20 | 3 | >80 | >16 | <0.1 | Plexiglass |
| [24] | 25.5–29.6 | Four/Connected | 30 × 35 | 8.3 | - | >15 | <0.01 | Rogers R04350B |
| [25] | 27.5–29 | Four/Connected | 25 × 15 | 7.8 | 95 | >17 | <0.0001 | Rogers 5880 |
| [26] | 25.1–37.5 | Four/Separate | 12.7 × 12 | 5 | 80 | >22 | <0.1 | Rogers 5880 |
| Proposed Three-Element | 26–40 | Three/Connected | 15 × 15 | >5 | >70 | >20 | <0.01 | Duroid5880 |
| Proposed Four-Element | 26–40 | Four/Connected | 15 × 15 | >7 | 60 | >20 | <0.02 | Duroid5880 |

## 7. Conclusions

Three-element (case 1) and four-elements (case 2) connected ground MIMO antennas are designed and developed in this article. First, a single antenna element is designed for the wideband frequency range from (26 GHz to 40 GHz) with a partial ground profile. Using the unit element, three-element and four-element MIMO antennas are designed, fabricated, and tested. The proposed design provides an impedance bandwidth of 43% (26 GHz to 40 GHz) for both cases. The placement of all the elements is decided such that it provides maximum isolation. In both cases, they achieve more than 20 dB isolation between the elements. Diversity analysis is conducted for the proposed design; the ECC values derived from the radiation pattern are less than 0.02 for both the antenna and the DG value is near 9.96. The efficiency of the antenna is more than 60% and the maximum achieved gain is 9 dBi, which makes the three-element and four-element MIMO antennas suitable for millimeter-wave 5G applications.

**Author Contributions:** Conceptualization, A.P., A.V. and I.E.; methodology, A.D., H.M. and A.P.; software, A.P. and I.E.; validation, C.Z. and D.C.; formal analysis, A.P.; investigation, A.V., K.M. and A.D.; resources, H.M.; writing—original draft preparation, A.P. and A.V.; writing—review and editing., K.M., I.E. and C.Z.; supervision, J.R. and A.D.; project administration, J.R.; funding acquisition, I.E. All authors have read and agreed to the published version of the manuscript.

**Funding:** This work is funded by FCT/MCTES through national funds and when applicable co-funded EU funds under the project UIDB/50008/2020-UIDP/50008/2020.

**Data Availability Statement:** All data are included within the manuscript.

**Acknowledgments:** This work is supported under the Charotar University of Science and Technology (CHARUSAT) Research Seed Grant scheme. The authors would like to thank CHARUSAT University for providing various resources for the successful completion of the work.

**Conflicts of Interest:** The authors declare no conflict of interest.

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
