# Peer review of "Inverted-L Shaped Wideband MIMO Antenna for Millimeter-Wave 5G Applications"

_electronics, doi:10.3390/electronics11091387_

Round 1

Reviewer 1 Report

This paper deals with a wideband MIMO antenna for the 26-40 GHz frequency band. Initially, a single element has been optimized for the wideband operating frequency. Use of the single element structure, 3- and 4-elements MIMO antennas are implemented. Both the MIMO antennas have connected to the ground plane met practical application. This reviewer thinks the design is very good; and suggests the manuscript can be accepted and to be published. 

Author Response

Original Manuscript ID: electronics-1701510

Original Article Title: “Inverted-L Shaped Wideband MIMO Antenna for mm-wave 5G Applications”

To: electronics

Re: Response to reviewers

Dear Editor,

Thank you for allowing a resubmission of our manuscript, with an opportunity to address the reviewers’ comments.

We are uploading (a) our point-by-point response to the comments (below) (response to reviewers), (b) an updated manuscript with yellow highlighting indicating changes, and (c) a clean updated manuscript without highlights (PDF main document).

Best regards,

Dr. Issa Elfergani

Reviewer #1

This paper deals with a wideband MIMO antenna for the 26-40 GHz frequency band. Initially, a single element has been optimized for the wideband operating frequency. Use of the single element structure, 3- and 4-elements MIMO antennas are implemented. Both the MIMO antennas have connected to the ground plane met practical applications. This reviewer thinks the design is very good; and suggests the manuscript can be accepted and to be published. 

Author Response: Authors are thankful to the reviewer for appreciating our work.  

Reviewer 2 Report

The authors have presented 3-element and 4-element MIMO antennas for mm-wave 5G applications. They have exhibited advantageous or comparable performances in table 1 to the state of art in literature. The paper shows wideband characteristics covering 26-40GHz, and the simulated and measured results validate the design. Some suggestions are listed below.

  1. There needs more discussion on the design of the unit antenna element. The readers will be interested in how the structure is designed and why it can work for wideband.
  2. For the comparison between simulation and measurement, the simulated curves can be plotted together with the measured in FIg. 14 and Fig. 15. 
  3. The element MIMO design for case 2 is obvious due to the symmetry and polarization orthogonality consideration. Figs. 6 and 7 and the related discussion are unnecessary. Please also explain why the slight asymmetry is intentionally (?) imposed in the design in Fig. 8 such that the four elements show slightly different performances.
  4. It is confusing to mention both diversity gain and gain. What is the major distinction? There is no reference for the definition of diversity gain. It is not an index of interest in the comparison table 1 either.
  5. The present design has the worst efficiency in comparison table 1. Please comment the reason why and suggest the improvement in design.

Author Response

Original Manuscript ID: electronics-1701510

Original Article Title: “Inverted-L Shaped Wideband MIMO Antenna for mm-wave 5G Applications”

To: electronics

Re: Response to reviewers

Dear Editor,

Thank you for allowing a resubmission of our manuscript, with an opportunity to address the reviewers’ comments.

We are uploading (a) our point-by-point response to the comments (below) (response to reviewers), (b) an updated manuscript with yellow highlighting indicating changes, and (c) a clean updated manuscript without highlights (PDF main document).

Best regards,

Dr. Issa Elfergani

Reviewer #2

Reviewer#2, Concern # 1: There needs more discussion on the design of the unit antenna element. The readers will be interested in how the structure is designed and why it can work for wideband.

Author Response: Thank you for your valuable suggestion. The authors have added a detailed explanation of the working of elementary antenna design in the wideband regime as below:

The result of the reflection coefficient value for the different ground planes is shown in Figure 2. The length of the ground plane is tuned from 7 mm to 15 mm (Full ground plane). The result indicates that a ground length of 11 mm provides a better reflection coefficient and impedance bandwidth in the operating range of (43%) 26 – 40 GHz compared to a full ground plane antenna that provides an impedance bandwidth of (33%) - 38 GHz.

Figure 2. Reflection coefficient values for the various ground plane

Author Action: The updated explanation is added to the last paragraph on page no. 3 which indicates the information about wideband response achieved using the proposed method.

Reviewer#2, Concern # 2: For the comparison between simulation and measurement, the simulated curves can be plotted together with the measured in Fig. 14 and Fig. 15. 

Author Response: Thank you for your valuable suggestion.

The figures are updated as shown below:

Figure 14.  Measured S-parameters result of the 3-element antenna (case 1).

Figure 15. Measured S-parameters result of the 4-element antenna (case 2)

Author Action: The updated figures are added in the revised manuscript.

Reviewer#2, Concern # 3: The element MIMO design for case 2 is obvious due to the symmetry and polarization orthogonality consideration. Figs. 6 and 7 and the related discussion are unnecessary. Please also explain why the slight asymmetry is intentionally (?) imposed in the design in Fig. 8 such that the four elements show slightly different performances.

Author Response: Thank you for your valuable suggestion. The authors would like to clarify the reason the for inclusion of Fig 6 (a-b). The effect on the mutual coupling due to the placement of elements plays an important role and so different configurations are simulated to give the readers a better idea about the placement of antenna elements in 4 element MIMO configuration.

The design in Fig 8 is chosen after carrying out various parametric analyses in terms of reflection coefficient and isolation among inter-element. It is ensured that even with such placement, all the 4 elements cover the wide band of interest.  

Reviewer#2, Concern # 4: It is confusing to mention both diversity gain and gain. What is the major distinction? There is no reference for the definition of diversity gain. It is not an index of interest in comparison to table 1 either.

Author Response: Thank you for your valuable query. The authors would like to clarify that gain indicates how strong a signal an antenna can send or receive in a specified direction whereas diversity gain is specifically used to determine the MIMO performance. This parameter is indicative of the reliability of the MIMO system. Isolation of the radiators is higher for high DG antenna systems

Reviewer#2, Concern # 5: The present design has the worst efficiency in comparison table 1. Please comment on the reason why and suggest the improvement in design.

Author Response: Thank you for your valuable query. The reduction in the efficiency is because of the wideband response of the proposed structure and the discontinuous ground plane in the design. The gain of the antenna could be improved by incorporating artificial magnetic conductors, array structures, frequency selective surfaces, and superstrate structures. 

Reviewer 3 Report

1- The language of the paper  should be revised

2- Add the application of the antenna please 

3- The S-parameters measured and simulated  results should be added on the same figure. 

4- The quality of figures should be improved

Author Response

Original Manuscript ID: electronics-1701510

Original Article Title: “Inverted-L Shaped Wideband MIMO Antenna for mm-wave 5G Applications”

To: electronics

Re: Response to reviewers

Dear Editor,

Thank you for allowing a resubmission of our manuscript, with an opportunity to address the reviewers’ comments.

We are uploading (a) our point-by-point response to the comments (below) (response to reviewers), (b) an updated manuscript with yellow highlighting indicating changes, and (c) a clean updated manuscript without highlights (PDF main document).

Best regards,

Dr. Issa Elfergani

Reviewer #3

Reviewer#3, Concern # 1: The language of the paper should be revised.

Author Response: The manuscript is checked thoroughly for typos and grammar mistakes and the same is corrected.

Author Action: The manuscript is updated accordingly and changes are highlighted and marked in red.

Reviewer#3, Concern # 2: Add the application of the antenna, please. 

Author Response: Millimeter-wave resonators are used in plenty of applications that include remote sensing, radio astronomy, automotive radars, military applications, mobile telecommunications,  remote sensing, security transmission, military applications, and imaging.

Author Action: The applications are added in the introduction section.

Reviewer#3, Concern # 3: The S-parameters measured and simulated results should be added on the same figure. 

Author Response: Thank you for your valuable suggestion.

The figures are updated as shown below:

Figure 14.  Measured S-parameters result of the 3-element antenna (case 1).

Figure 15. Measured S-parameters result of the 4-element antenna (case 2)

Author Action: The updated figures are added in the revised manuscript.

Reviewer#3, Concern # 4: The quality of figures should be improved.

Author Response: Thank you for your suggestion. The figures are improved for better clarity.

Author Action: The updated figures are added in the revised manuscript.

Reviewer 4 Report

The work presented is interesting and there are not much comments except literature which is almost ignored. I am pointing out of comments here,

  1. Many grammatical mistakes found, authors should revise it carefully
  2. Abstract is not written well, it is recommended to be consistent.
  3. There is almost no past study added, I strongly suggest to add literature (around 10), I am copying here some,
  • https://doi.org/10.3390/app12073684
  • https://doi.org/10.3390/app11188331   
  1. What are the difference between both designs?
  2. What is meant by ‘MIMO Diversity’?
  3. Authors should add a column in table.1 to show the difference with already published work

Author Response

Original Manuscript ID: electronics-1701510

Original Article Title: “Inverted-L Shaped Wideband MIMO Antenna for mm-wave 5G Applications”

To: electronics

Re: Response to reviewers

Dear Editor,

Thank you for allowing a resubmission of our manuscript, with an opportunity to address the reviewers’ comments.

We are uploading (a) our point-by-point response to the comments (below) (response to reviewers), (b) an updated manuscript with yellow highlighting indicating changes, and (c) a clean updated manuscript without highlights (PDF main document).

Best regards,

Dr. Issa Elfergani

Reviewer #4

Reviewer#4, Concern # 1: Many grammatical mistakes found, authors should revise it carefully.

Author Response: Thank you for your valuable suggestion. The manuscript is checked thoroughly for typos and grammar mistakes and the same is corrected.

Author Action: The manuscript is updated accordingly and changes are highlighted and marked in red.

Reviewer#4, Concern # 2: Abstract is not written well, it is recommended to be consistent.

Author Response: Thank you for your valuable suggestion. The abstract is updated as below:

An interconnected three-elements and four-elements wideband MIMO antennas have been proposed for mm-wave-5G applications by performing numerical computations and carrying out experimental measurements. The antenna structure is realized using Rogers 5880 substrate (εr=2.2, tan δ=0.0009), where the radiating element has a shape of inverted-L with the partial ground. The unit element is carefully designed and positioned (by orthogonally rotating the elements) to form 3-elements (case 1) and 4-element (case 2) MIMO antennas. The interconnected ground for both cases is ascertained to increase the practical utilization of the resonator. The proposed MIMO antenna size for case 1 is (0.95l × 3l) and (2.01l × 1.95l) for case 2 (at the lowest functional frequency). Both the designs give the impedance bandwidth of approximately 26-40 GHz (43 %). Moreover, they achieve greater than 15 dB isolation, more than 6 dBi gain with an ECC value lower than 0.02 which meets the MIMO diversity performance thus making the 3-elements and 4-elements MIMO antennas the best choice for mm-wave 5G applications.

Author Action: The abstract is updated in the revised manuscript.

Reviewer#4, Concern # 3: There is almost no past study added, I strongly suggest to add literature (around 10), I am copying here some,

  • https://doi.org/10.3390/app12073684
  • https://doi.org/10.3390/app11188331   

Author Response: Thank you for the valuable suggestion. The literature is enhanced by adding the references as shown below:

4-element wideband MIMO antenna for the 9.33 GHz of the center frequency is proposed in [28]. It covers the frequency range from 2.77 GHz to 12 GHz. A fractal singular ring is used as the front side of the antenna and trapezoidal partial ground is at the bottom plane of the antenna. In [29] MIMO antenna for 5G handheld devices is proposed and implemented. Total 6 antennas are part of the design out of which 2 are working on sub 6GHz band and four are working on mm-wave 5G band. A dual-band antenna with 2-element for the operating frequency of 2.4 GHz and 5.8 GHz is proposed in [30]. For the improvement of the isolation between the elements adopting parasitic elements and the defected ground plane is used. Tree shape planar MIMO antenna for the 5G mm-wave application is proposed in [31]. To achieve wideband response radiating elements with the various arc is used. A small slot MIMO antenna that can be embedded in the smart glasses is designed with the experimental validation in [32].   

[28] Alharbi, Abdullah G., UmairRafique, ShakirUllah, Salahuddin Khan, Syed Muzahir Abbas, EsraaMousa Ali, Mohammad Alibakhshikenari, and Mariana Dalarsson. "Novel MIMO Antenna System for Ultra-Wideband Applications." Applied Sciences 12, no. 7 (2022): 3684.

[29] Khalid, Hassan, Wahaj Abbas Awan, Musa Hussain, Adeela Fatima, Mudassir Ali, NiamatHussain, Salahuddin Khan, Mohammad Alibakhshikenari, and Ernesto Limiti. "Design of an Integrated Sub-6 GHz and mmWave MIMO Antenna for 5G Handheld Devices." Applied Sciences 11, no. 18 (2021): 8331.

[30] Peng, He, RuixingZhi, Qichao Yang, Jing Cai, Yi Wan, and Gui Liu. "Design of a MIMO Antenna with High Gain and Enhanced Isolation for WLAN Applications." Electronics 10, no. 14 (2021): 1659.

[31] Sehrai, Daniyal Ali, Mujeeb Abdullah, Ahsan Altaf, Saad Hassan Kiani, Fazal Muhammad, Muhammad Tufail, Muhammad Irfan, Adam Glowacz, and Saifur Rahman. "A novel high gain wideband MIMO antenna for 5G millimeter wave applications." Electronics 9, no. 6 (2020): 1031.

[32] Chung, Ming-An, Cheng-Wei Hsiao, Chih-Wei Yang, and Bing-Ruei Chuang. "4× 4 MIMO Antenna System for Smart Eyewear in Wi-Fi 5G and Wi-Fi 6e Wireless Communication Applications." Electronics 10, no. 23 (2021): 2936.

Author Action: The abstract is updated in the revised manuscript.

Reviewer#4, Concern # 4: What are the differences between both designs?

Author Response: Thank you for the valuable query. The authors would like to emphasize that first a single element antenna is optimized. Later the single element is arranged in a horizontal fashion to realize a 3-element connected ground structure. To accommodate a greater number of users, the single element antenna is arranged in an asymmetric rotational fashion ensuring that the required band of interest is achieved while maintaining the required isolation of more than 15 dB between the antenna elements.   

Reviewer#4, Concern # 5: What is meant by ‘MIMO Diversity’?

Author Response: Thank you for the valuable query. Diversity analysis is a must when proposing a MIMO antenna along with other fundamental parameters. The diversity parameters include mutual coupling and isolation, envelope correlation coefficient (ECC), diversity gain (DG), etc. In applications using MIMO antennas, multiple antennas transmit the signals and as a result, it is expected that those are uncorrelated and independent.

Reviewer#4, Concern # 6: Authors should add a column in table.1 to show the difference with already published work.

Author Response: Thank you for the valuable suggestion. The authors have added a paragraph showing how the proposed antenna is better than other states of the arts:

From Table 1, it is observed that [20-21] provides single-frequency operation whereas the proposed antenna provides wideband frequency response with compact size, better ECC, and gain value. The antenna proposed in [22-23] provides the dual-frequency response which is overcome in the proposed design as it operates in the wideband regime with a compact size. The size of the proposed design is small as compared to [24-25] with wider frequency response and better isolation between the ports.  MIMO antenna in [26] has a compact size compared to the proposed antenna; however, the separate ground structure makes the practical utilization of the antenna limited.   
